# Optimising paediatric afferent component early warning systems: a hermeneutic systematic literature review and model development

Nina Jacob ,[1] Yvonne Moriarty ,[1] Amy Lloyd,[1] Mala Mann,[2] Lyvonne N Tume,[3] Gerri Sefton,[4] Colin Powell,[5,6] Damian Roland,[7,8] Robert Trubey,[1] Kerenza Hood,[1] Davina Allen[9]

For numbered affiliations see end of article.

**Correspondence to**
Dr Nina Jacob;
Jacobn@cardiff.ac.uk

## ABSTRACT

**Objective** To identify the core components of successful early warning systems for detecting and initiating action in response to clinical deterioration in paediatric inpatients.

**Methods** A hermeneutic systematic literature review informed by translational mobilisation theory and normalisation process theory was used to synthesise 82 studies of paediatric and adult early warning systems and interventions to support the detection of clinical deterioration and escalation of care. This method, which is designed to develop understanding, enabled the development of a propositional model of an optimal afferent component early warning system.

**Results** Detecting deterioration and initiating action in response to clinical deterioration in paediatric inpatients involves several challenges, and the potential failure points in early warning systems are well documented. Track and trigger tools (TTT) are commonly used and have value in supporting key mechanisms of action but depend on certain preconditions for successful integration into practice. Several supplementary interventions have been proposed to improve the effectiveness of early warning systems but there is limited evidence to recommend their wider use, due to the weight and quality of the evidence; the extent to which systems are conditioned by the local clinical context; and the need to attend to system component relationships, which do not work in isolation. While it was not possible to make empirical recommendations for practice, the review methodology generated theoretical inferences about the core components of an optimal system for early warning systems. These are presented as a propositional model conceptualised as three subsystems: detection, planning and action.

**Conclusions** There is a growing consensus of the need to think beyond TTTs in improving action to detect and respond to clinical deterioration. Clinical teams wishing to improve early warning systems can use the model to consider systematically the constellation of factors necessary to support detection, planning and action and consider how these arrangements can be implemented in their local context.

**PROSPERO registration number** CRD42015015326.

### Strengths and limitations of this study

► The literature in this field is heterogeneous and better at identifying system weakness than it is effective improvement interventions. By deploying social theories and a hermeneutic review methodology it was possible to develop a propositional model of the core components of an afferent component paediatric early warning system.

► The model is derived from logical inferences drawing on the overall evidence synthesis, social theories and clinical expertise, rather than strong empirical evidence of single intervention effectiveness.

► There is a growing consensus of the need to take a whole systems approach to improve the detection and response to deterioration in the inpatient paediatric population and this paper offers an evidence-based framework for this purpose.

## INTRODUCTION

Failure to recognise and act on signs of clinical deterioration in the hospitalised child is an acknowledged safety concern.[1] Track and trigger tools (TTT) are a common response to this problem. A TTT consists of sequential recording and monitoring of physiological, clinical and observational data. When a certain score or trigger is reached then a clinical action should occur including, but not limited to, altered frequency of observation, senior review or more appropriate treatment or management. Tools may be paper based or electronic and monitoring can be automated or undertaken manually by staff.

Despite the growing use of TTTs there is limited evidence of their effectiveness as a single intervention in reducing mortality or arrest rates in hospitalised children.[2] [3] Results from the largest international cluster randomised controlled trial of a TTT (the Bedside Paediatric Early Warning System

(BedsidePEWS)) did not support TTT use to reduce mortality, and highlighted the multifactorial mechanisms involved in detecting and initiating action in response to deterioration.[4] These findings lend further weight to a developing consensus about the need to look beyond TTTs to the impact of wider system factors on detecting and responding to deterioration in the inpatient paediatric population.[2 5–9] This paper reports on a theoretically informed systematic hermeneutic literature review[10] to identify the core components and mechanisms of action of successful afferent component early warning systems (EWS) in paediatric hospitals and is one of three linked reviews undertaken as part of a wider UK study commissioned to develop and evaluate an evidence-based paediatric warning system.[3 11] It addressed the following question:

*What sociomaterial and contextual factors are associated with successful or unsuccessful Paediatric Early Warning Systems (with or without TTTs)?*

## METHOD
### Design
We performed a hermeneutic systematic review of the relevant literature. A hermeneutic systematic review is an iterative process, integrating analysis and interpretation of evidence with literature searching and is designed to develop a better understanding of the field.[10] The popularity of the method is growing in health services research where it has value in generating insights from heterogeneous literatures that cannot be synthesised through standard review methodology[12] and would otherwise produce inconclusive findings (see ref 9). The purpose of the review was not exhaustive aggregation of evidence, but to develop an understanding of the social, material and contextual factors associated with successful or unsuccessful paediatric early warning systems (PEWS).

### Theoretical framework
Data extraction and interpretation was informed by translational mobilisation theory (TMT)[13 14] and normalisation process theory (NPT).[15 16] TMT is a practice theory which explains how goal-oriented collaborative activity is mobilised in unpredictable environments (box 1) and how the relevant mechanisms of action are conditioned by the local context. It is well suited for understanding EWS which require the organisation of action in evolving conditions, in a variety of clinical environments, with different teams, skill mixes, resources, structures and technologies. NPT shares the same domain assumptions as TMT and is concerned with 'how and why things become, or do not become, routine and normal components of everyday work',[15 16] directing attention to the preconditions necessary for successful implementation of interventions. The theoretical framework informed our data extraction strategy, interpretation of the evidence and the development of a propositional model of an optimal paediatric early warning system.

---

**Box 1 Mechanisms of translational mobilisation and their application to rescue trajectories[14]**

*Object formation*—how people draw on the interpretative resources available to them within a strategic action field to create the objects of their practice. Enrolment into an escalation trajectory requires multiple examples of object formation beginning with construction of an individual as at risk of deterioration and a regime of vital signs monitoring instigated, through recognition that the patient's physiological status is a cause for concern, to the identification of the patient as requiring a specific intervention. How this is achieved is highly dependent on the features of the local strategic action field.

*Translation*—the processes that enable practice objects to be shared and different understandings accommodated. It points to the actions necessary in order for a patient that is an object of concern for nursing staff to be translated into a clinical priority for the doctor and, if necessary, to be translated into the focus of intervention by the emergency response team.

*Articulation* refers to the secondary work processes that align the actions, knowledge and resources necessary for the mobilisation of projects of collective action. It is the work that makes the work, work. Responding to deterioration is time critical and articulation work is necessary to ensure the availability of resources and materials to support clinical management. This is not a mundane observation; catastrophic failures in patient safety are often attributed to the lack of functioning equipment[107] and the absence of monitoring equipment has been identified as a factor undermining the implementation of early warning track and trigger tools.[48] Attending to articulation in rescue trajectories also underlines the temporal ordering of action and the mechanisms required to achieve this, directing improvement efforts towards the organisation's escalation policy, for example.

*Reflexive monitoring* refers to the processes through which people collectively or individually appraise and review activity. In a distributed field of action, reflexive monitoring is the means through which members accomplish situational awareness[108] of an overall project. The importance of situation awareness in rescue trajectories is well recognised, but achieving this is challenging. Reflexive monitoring is conditioned by the wider institutional context which will include a multiplicity of informal and formal mechanisms designed for this purpose: nursing and medical handovers, the ward round, safety briefings. The form, frequency and effectiveness of these processes in supporting detecting and acting on deterioration would need to be taken into account in any improvement initiative.

*Sensemaking* refers to the processes through which agents create order in conditions of complexity. It draws attention to how the material and discursive processes by which members organise their work, account for their actions and construct the objects of their practice also give meaning and substance to the institutional components of strategic action fields that shape activity and condition future activity.

---

### Focus of the review
The literature in this field identifies four integrated components which work together to provide a safety system for at-risk patients: (1) the afferent component which detects deterioration and triggers timely and appropriate action; (2) the efferent component which consists of the people and resources providing a response; (3) a process improvement component, which includes system auditing and monitoring; and (4) an administrative component focusing on organisational leadership

and education required to implement and sustain the system.[17] Our focus was limited to the afferent components of the system.

## Stages of the review

### Stage 1: scoping the literature

Literature was identified through a recent scoping review,[7] team members' knowledge of the field, hand searches and snowball sampling techniques. The purpose was to (1) inform our review question and eligibility criteria and (2) identify emerging themes and issues. While we drew on several reviews of the literature[5 12–14] we always consulted original papers. Data were extracted using data extraction template 1 (online supplementary appendix 1) and analysed to produce a provisional conceptual model of the core components of paediatric early warning systems. Additional themes of relevance were identified: family involvement, situational awareness (SA), structured handover, observations and monitoring and the impact of electronic systems and new technologies.

### Stage 2: searching for the evidence

We undertook systematic searches of the paediatric and adult EWS literature (the goals and mechanisms of collective action in detection and rescue trajectories are the same). For the adult literature we used the same search strategies but added a qualitative filter to limit the scope to studies most likely to yield the level of sociomaterial and contextual detail of value to the review. Literature informing additional areas of interest was located through a combination of systematic and hand searches. Systematic searches (searches 2 and 3) were undertaken in areas where we anticipated locating evidence of the effectiveness of specific interventions to strengthen EWS. Theory-driven searches reflected the conceptual requirements of the model development.

#### Systematic searches

A systematic search was initially conducted across a range of databases from 1995 to September 2016 to identify relevant studies on the PEWS literature. This search was updated to cover literature from September 2016 to May 2018. An additional three systematic searches were conducted from 1995 to September 2016 to identify supplementary papers to aid in developing understanding on the PEWS literature:

1. Adult EWS.
2. Interventions to improve SA.
3. Structured communication tools for handover and handoff.

Detailed information on the search methodology can be found in online supplementary appendix 2. Grey literature was excluded in order to keep the review manageable.

#### Theory-driven searches

Additional theory-driven searches were conducted in the following areas:

1. Family involvement.

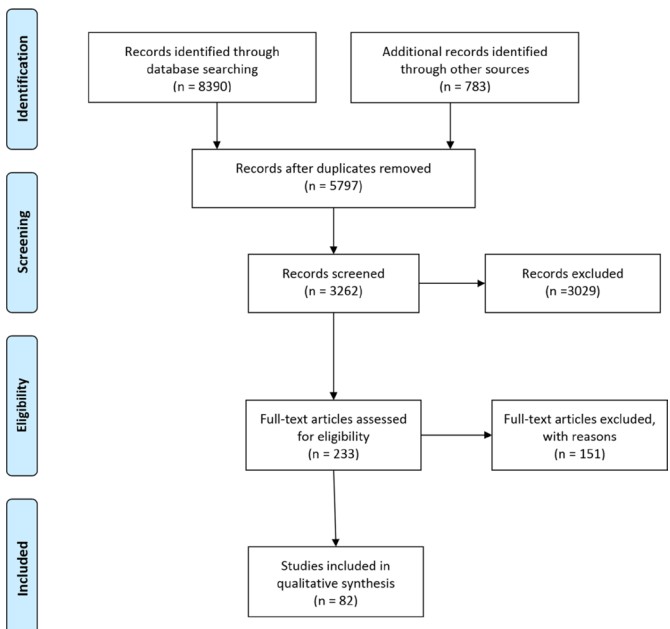

**Figure 1** Preferred Reporting Items for Systematic Reviews and Meta-Analyses (PRISMA) diagram (adapted from Moher et al[109]).

2. Observations and monitoring.
3. The impact of electronic systems.

These were a combination of exploratory, computerised, snowball and hand searches. As the analysis progressed, we continued to review new literature on EWS as this was published.

#### Screening

After removing duplicates 5284 references were identified for screening. A modified Preferred Reporting Items for Systematic Reviews and Meta-Analyses flow diagram is provided (figure 1). Papers were screened by title to assess eligibility and then by full text to assess relevance for data extraction. The PEWS and adult EWS searches were screened by two researchers, searches 2 and 3 were screened by the lead reviewer.

### Stage 3: data extraction and appraisal

Data extraction template 2 (online supplementary appendix 3) was applied to all papers included in the review. As is typical of reviews of this kind, evidential fragments and partial lines of inquiry formed the unit of analysis rather than whole papers.[18] These fragments were quality assessed according to the contribution they made to the developing analysis rather than assessing the paper as whole through the use of formal appraisal tools. Data extraction and quality appraisal were undertaken concurrently and double checked by a second reviewer.

### Stage 4: developing a propositional model

A propositional model was developed specifying the core ingredients of a paediatric early warning system (table 1). It comprises logical inferences derived from the theoretical framework and evidence synthesis, informed by

| Table 1 | Propositional model | | |
|---|---|---|---|
| | **Proposition** | **Conceptual requirements** | |
| Detection | Detection of deterioration depends on timely and appropriate *monitoring* of vital signs and relevant risk factors. | At a minimum, this requires:<br>▶ Staff are aware of which vital signs need to be monitored.<br>▶ Staff are aware of the minimum frequency of observations required for the children in their care.<br>▶ Staff are aware of the need to review the frequency of observations for children in their care.<br>▶ Staff are aware of additional clinical assessments required for children with prior risk factors.<br>▶ Monitoring tasks are allocated to staff members with appropriate skills to conduct them.<br>▶ Staff have access to appropriate equipment to accurately monitor vital signs, and conduct other clinical assessments.<br>▶ Staff are aware of roles and responsibilities for monitoring.<br>▶ Staff have time to conduct accurate, timely and appropriate monitoring of vital signs, alongside other work commitments.<br>▶ Staff concern is formally recognised as a valid indicator of deterioration.<br>▶ Staff are supported to develop and use their intuition in detecting signs of deterioration.<br>▶ Staff understand the value of family concerns in the detection of deterioration.<br>▶ Families are involved with defining normal physiological parameters for their child.<br>▶ Families receive guidance about what to do if they are concerned that their child's condition is deteriorating.<br>▶ Staff keep families informed about developments in their child's care and treatment. | |
| Detection | Detection of deterioration depends on timely and appropriate *recording* of signs of deterioration. | At a minimum this requires:<br>▶ Staff are aware of the need to record vital signs, family concern and staff concern promptly and accurately.<br>▶ Staff are aware of roles and responsibilities for recording vital signs, family concern and staff concern.<br>▶ Staff have appropriate skills to accurately record vital signs, family concern and staff concern.<br>▶ Staff have access to appropriate equipment to accurately record vital signs, family concern and staff concern.<br>▶ There are an appropriate number of staff to carry out required tasks. | |
| Detection | Detection of deterioration depends on timely and appropriate *interpretation* of signs of deterioration. | At a minimum this requires:<br>▶ Staff are aware of prior factors that increase children's risk of deterioration (eg, premature birth).<br>▶ Staff are aware of roles and responsibilities for interpreting signs of deterioration.<br>▶ Staff take into account vital signs, family concern and staff concern in assessing the condition of children in their care.<br>▶ Teams have appropriate skills to discern patterns and trends of signs and symptoms.<br>▶ Staff have the opportunity to learn how to interpret signs of deterioration from shadowing more senior staff.<br>▶ Care is organised to enable staff to recognise patterns and trends for children.<br>▶ Families are in a position to discern patterns of signs and symptoms in their child. | |
| Planning | Planning depends on *reviewing* indicators of deterioration for each patient. | At a minimum this requires:<br>▶ For each child, all indicators of deterioration are brought together and kept up to date.<br>▶ There is a regular mechanism for reviewing the status of all children in the ward to identify those children who are a concern.<br>▶ There is a regular mechanism for reviewing staffing levels and skills mix, workload, acuity and admissions. | |

Continued

**Table 1** Continued

| | Proposition | Conceptual requirements |
|---|---|---|
| Planning | Planning depends on staff being aware at ward level of the status of individual patients and the availability of skills and resources, and *preparing* an appropriate response. | At a minimum this requires: <br>▶ There is a regular mechanism for communicating the review of all children, staffing levels and other resources to the rest of the team and senior managers. <br>▶ There is a regular mechanism for planning appropriate response to deterioration. <br>▶ Senior staff members are allocated responsibility for managing demand and resources. <br>▶ Senior staff members are allocated responsibility for communicating response plans. <br>▶ There is an action plan for children at risk of deterioration which is shared with families and staff caring for them. |
| Action | Action depends on clear escalation and response processes. | At a minimum this requires: <br>▶ A trigger or prompt to act from detection or planning phases. <br>▶ Clearly defined graded escalation and response procedures—agreed at organisational level. <br>▶ Staff receive guidance about how to escalate and respond. <br>▶ Staff understand their roles and responsibilities in the escalation procedure as activators and responders. <br>▶ Staff are encouraged and supported in raising concerns. <br>▶ Families are encouraged and supported in raising concerns. <br>▶ Staff are able to communicate information across professional hierarchies using a structured approach to sharing information. <br>▶ Clear structures to support action, including the use of a 'no false alarms' policy so staff are not deterred from escalating care. |
| Action | Action depends on evaluation. | At a minimum this requires: <br>▶ Escalation and response processes are reviewed to promote learning. <br>▶ There is opportunity for staff to discuss differences of opinion in the need for escalation. <br>▶ No blame is assigned to those who escalate. |

clinical experts on the team. Iterations of the model were developed in collaboration with clinical colleagues. A series of face-to-face meetings were conducted to review structure, wording and applicability to clinical practice.

### Patient and public involvement

This review was conducted as part of a larger mixed methods study (ISRCTN 94228292), which used a formal, facilitated parental advisory group. The group comprised parents of children who had experienced an unexpected adverse event in a paediatric unit and provided input which helped shape the broader research questions and wider contextual factors to consider, specifically within the family involvement element of the system. The results of the review will be disseminated to parents through this group.

### RESULTS
### Included studies

Eighty-two papers were included in the review. Forty-six papers focused on TTT implementation and use in paediatric and adult contexts (24 from the paediatric search and the remaining 22 from the adult-focused search); the remaining 36 papers contributed supplementary data on factors related to the wider warning system. See table 2 for a detailed breakdown of this process. No studies were located that adopted a whole systems approach to detecting and responding to deterioration.

### Analysis

In TMT the primary unit of analysis is the 'project', which defines the social and material actors (people, materials, technologies) and their relationships involved in achieving a particular goal. The goals of the afferent paediatric warning system are: first, that the child is identified as at risk and a vital signs monitoring regime instigated; second, that evidence of deterioration is identified through monitoring and categorised as such; and third, that timely and appropriate action is initiated in response to deterioration. Our analysis of the literature suggests that three subsystems within the afferent component of EWS support these processes: the *detection* of signs deterioration; the *planning* needed to ensure teams are ready to act when deterioration is detected; and the initiation of timely *action*. While we have focused on the afferent component, it is important to remember that all elements of the overall safety system (efferent component, process improvement and administrative arm) need to be working in concert in order to maintain an optimal paediatric early warning system. In the next section, we report on the literature in relation to each subsystem.

**Table 2** Screening breakdown

| | PEWS | EWS | Structured handover | Situational awareness | Electronic systems | Observations and monitoring | Family involvement | Snowball sample |
|---|---|---|---|---|---|---|---|---|
| Database searching | 3564 | 1155 | 3369 | 302 | – | – | – | – |
| Additional sources | 83 | 7 | 150 | 46 | – | – | – | – |
| Records after duplicates removed | 2194 | 751 | 2156 | 199 | – | – | – | – |
| Hand searches | 431 | – | – | – | 26 | 20 | 15 | 5 |
| Title screening | 90 | 751 | 2156 | 199 | 26 | 20 | 15 | 5 |
| Abstract screening | 62 | 106 | N/A | N/A | 26 | 20 | 15 | 5 |
| Full paper screening | 39 | 65 | 37 | 26 | 26 | 20 | 15 | 5 |
| *Included in syntheses* | *24* | *22* | *4* | *6* | *10* | *2* | *9* | *5* |

EWS, early warning system; N/A, not applicable; PEWS, Paediatric Early Warning System.

## Detection

The goal of the detection subsystem is to recognise early signs of deterioration, so the child becomes the focus of further clinical attention. This requires, first, that the child is identified as at risk and a vital signs monitoring regime instigated and, second, that the child is identified as showing signs of deterioration.

Despite widespread use, the evidence on TTT effectiveness in predicting adverse outcomes in hospitalised children is weak.[3] Many TTTs have only been validated retrospectively and postpredictive values were generally low. Studies reporting significant decreases in cardiac arrest calls or mortality had methodological concerns. The literature does suggest that TTTs have value in supporting process mechanisms in the detection subsystem. Vital signs monitoring is undertaken on all hospital inpatients and, like other high-volume routine activity, is often delegated to junior staff[19–38] who may not have sufficient skills to interpret results.[21 22 37] TTTs have value in mitigating these risks: by specifying physiological thresholds that indicate deterioration they take knowledge to the bedside and act as prompts to action[19 39] which can lead to a more systematic and frequent approach to monitoring and improved detection of deterioration.[40 41]

TTT's effectiveness in fulfilling these functions depends on certain preconditions. The review highlighted that TTT use was impacted by the availability of appropriate and functioning equipment,[22 27 29 34 39 42–46] (in)adequate staffing and night-time pressures[22 26 29 30 37 40 42–44 47–52] and an appropriately skilled workforce.[26 28 36 43 49 50 53–57] On this latter point, while several papers report on education packages to improve the detection of deterioration, the evidence is not robust enough to recommend specific programmes.[23–25 28 30 35 55 58–60] There were also times whereby nursing staff prioritised sleep over waking a patient to take vital signs.[46 61]

TTTs are also used differently depending on the experience of the user. For juniors, they provide a methodology and structure for monitoring clinical instability and identifying deterioration, whereas more experienced staff reportedly use TTTs as confirmatory technologies.[19–32] The importance of professional intuition in detecting deterioration is extensively reported across the literature[19–22 26 27 29 31 32 36–40 42 43 45 46 48 50–53 60 62–68] and several authors recommend the inclusion of 'staff concern' in tool criteria.[26 48 51 57] This is important; TTTs may be of less value in patients with chronic conditions because of altered normal physiology or where subtle changes are difficult to detect.[64] It is also the case that TTTs are implemented in contexts governed by competing organisational logics which impact on their value and use.[43 54 58] For example, Mohammed Iddrisu *et al*[57] show TTTs have limited value immediately after surgery because acceptable vital sign parameters are different in the immediate postoperative period.

There is growing interest in the literature in strategies that facilitate patient and relative involvement in the early detection of deterioration.[69 70] Healthcare professionals depend on families to explain their child's normal physiological baseline and identify subtle changes in their child's condition but this information is not always systematically obtained.[71 72] Some authors propose family involvement in interdisciplinary rounds (This is an editorial paper),[73] but this requires parents to have detailed information about the signs and symptoms they should be attending to[72] and as yet there is little evidence on

effective strategies for how they might be involved in the detection of deterioration.[73]

While much of the literature reports on intermittent manual vital signs monitoring and paper-based recording systems, across the developed world there is a growing use of electronic technologies, which have important implications for the wider detection subsystem.[74] We considered a number of evaluations of new technologies which indicated that electronic vital signs recording is associated with a number of positive outcomes, particularly timeliness and accuracy, when compared with paper-based systems.[75 76] They can provide prompts or alerts for monitoring,[77–79] which facilitates better recognition of deterioration and is associated with a reduction in mortality.[78 80] These studies tend to evaluate new technologies in isolation, however, and do not engage with the literature highlighting alarm fatigue which is known to mitigate effectiveness over time or concerns about over-burdening staff with alerts.[81–83] Moreover, the successful implementation of new technologies is conditioned by the local context. For instance, where manual input into an electronic device is required, access to computers is an essential precondition. When computers were not available, staff 'batch' the collection of vital signs before data entry, thereby delaying the timely detection of deterioration.[27 45 84] In another study where the electronic system was found to be cumbersome and separated the collection and entry of data from the review of vital signs, verbal reports were favoured to ensure timely communication of information.[85] See table 3 for a summary of the evidence reported.

## Planning

Detecting and responding to deterioration involves the coordination of action in conditions of uncertainty and competing priorities. The goal of the 'Planning' subsystem is to ensure the clinical team are ready to act in the event of evidence of deterioration and is reflected in the growing interest in the literature on structures to facilitate team SA, group decisions and planning.[62]

TTTs have been found to support SA. Their use enabled clinicians to have a 'bird's-eye' view over all admitted patients on a ward as well as encouraging staff to consider projected acuity levels of the ward.[86] A number of studies also report on 'huddles' in facilitating SA.[32 65 87 88] A huddle is a multidisciplinary event scheduled at predetermined times where members discuss specific risk factors around deterioration and develop mitigation plans. One study combined the introduction of huddles with a 'watchstander', a role fulfilled by a charge nurse or senior resident, whose primary function is to know patients at high risk for deterioration.[88] These initiatives were associated with a near 50% reduction in transfers from acute to intensive care determined to be unrecognised situation awareness events. A further strategy identified by Goldenhar *et al* describes the use of the 'watcher' category to designate a patient as at risk where staff have a 'gut feeling' deterioration is likely.[87] A recent study used the category of 'watcher' to create a bundle of expectations to standardise communication and contingency planning. Once a patient was labelled 'a watcher' a series of five specific tasks, such as documentation of physician awareness of watcher status and that the family had been notified of the change in the patient's status, needed to be completed within 2 hours.[89]

Handovers are integral to clinical communication and contribute to SA. The extensive literature on handover indicates that information sharing can be of variable quality[47 54 90] and there is growing evidence that structured approaches improve this.[30 47 54 63 87 90–94] Ranging from a checklist system[91 93] to a cognitive aid developed through consensus,[23 94] most of the published interventions are variations of the Situation-Background-Assessment-Recommendation (SBAR) tool.[54 90–92] While effective handover depends on communicative forms that extend beyond the information transfer that is typically the focus of structured handover tools,[90] in the context of EWS a lack of standardisation allows greater margin for individualistic practices and difficulties accessing complementary knowledge and establishing shared understandings.[47]

There is also a literature on the use of common information spaces—such as whiteboards—in facilitating SA in the healthcare team.[23 33 47 53 55 58 67] These should be in a visible location and colour coded to correspond with the TTT score, where relevant.[47 55 58] Electronic systems automate this information and allow information to be reviewed remotely. However, they disconnect vital signs data from the patient and hence other indicators of clinical status and access to data is contingent upon the availability of computers.[27 45 47 84 95]

The literature indicates that SA can be facilitated in different ways in different contexts and it is the relationship between system elements that is important.[47] In their study on SA in delivery suites, Mackintosh *et al* discuss the three main supports for SA—whiteboard, handover and coordinator role—and illustrate how these interacted in organisations with strong SA compared with those with reduced levels. Crucially, this 'interplay' between the different activities was highly context dependent; 'the same supports used differently generate different outcomes' (p 52).[47] See table 4 for a summary of the planning evidence.

## Action

The goal of the 'Action' subsystem is to initiate appropriate action in response to evidence of deterioration. The literature suggests that mobilising action across professional boundaries/hierarchies is challenging, with differences in language between doctors and nurses and power dynamics contributory factors.[27 40 41 50 52 57 60 96] TTTs are in part a response to the challenges of communication in mobilising action in response to deterioration. By transforming a series of discrete observations into a summative indicator of deterioration—such as a score or a trigger—TTTs 'translate' and package the patient's status into a form that can be readily communicated

**Table 3** Summary of detection evidence

| Author | Country | Methodology | Analysis | Search area | Evidence contribution |
|---|---|---|---|---|---|
| Andrews and Waterman[19] | UK | Interviews and observations | Grounded theory | EWS | Importance of 'gut feeling' in detecting deterioration. Vital signs monitoring delegated to junior staff. |
| Astroth et al[50] | USA | Semistructured interviews with nurses | Coding categories were generated from the data, and consensus on final themes was achieved through an iterative process. | EWS | Staff encouraged to use their intuition when activating the RRT |
| Azzopardi et al[20] | Australia | Survey | Statistical analysis | PEWS | Track and trigger tools used to confirm or identify deterioration depending on experience. Useful for junior staff. Vital signs monitoring delegated to junior staff. |
| Bellomo et al[79] | International: USA, Sweden, UK, Netherlands, Australia | Before-and-after multicentred international controlled trial | Automated vital signs monitoring and early warning score calculated, international study, blinded trial, hospitals retained own early warning score and escalation policy. | Electronic systems | The electronic health record (EHR) provides a prompt to action. |
| Bonafide et al[21] | USA | Semistructured interviews | Grounded theory | PEWS | Vital signs monitoring delegated to junior workers who may not have the knowledge to interpret results. Track and trigger tools used to confirm or identify deterioration depending on experience and particularly useful for junior staff. Professional intuition important for senior staff to detect deterioration. |
| Bonafide et al[81] | USA | Prospective feasibility study | Video recording and electronic patient data collected prospectively. Pragmatic observational study of VitalPAC deployment in two large hospitals. | Electronic systems | Alarm fatigue—a barrier to escalation |
| Bonafide et al[83] | USA | Video review and response time outcome | Statistical analysis | PEWS | Alarm fatigue—a barrier to escalation |

Continued

**Table 3** Continued

| Author | Country | Methodology | Analysis | Search area | Evidence contribution |
|---|---|---|---|---|---|
| Braaten[22] | USA | Document review and interviews using the principles of cognitive work analysis | Inductive and deductive forms of analysis—cognitive work analysis, framework and directed content analysis | EWS | Vital signs monitoring delegated to junior workers who may not have the knowledge to interpret results. Track and trigger tools used to confirm or identify deterioration depending on experience. Professional intuition important for more senior staff/when clinical change is abrupt. Issues around availability of equipment and staffing. |
| Brady and Goldenhar[63] | USA | Focus groups ×7—held in groups of participants with similar roles | Constant comparison | Situational awareness | Paediatric early warning score supplementing gut feeling but these were not standardised. |
| Burns et al[68] | USA | Semistructured interviews were used drawing on a descriptive phenomenological methodology. | Iterative thematic analysis | Snowball sample | Importance of professional intuition is reported. |
| Chua et al[36] | Singapore | A qualitative survey using critical incident technique | Inductive content analysis | EWS | Vital signs monitoring is the responsibility of nurses. The regularity of this activity can lead to it being viewed as a mundane activity. Importance of 'gut feeling' and intuition in detection of deterioration. |
| Cioffi[48] | Australia | Unstructured interviews with nurses who had activated the medical emergency team (MET) | Simple code and retrieve | EWS | Importance of gut feeling and 'knowing' your patient in the detection of deterioration. Importance of having staff concern in the MET criterion. |
| Cioffi[51] | Australia | Unstructured interviews | Simple code and retrieve | EWS | Importance of gut feeling and intuition in recognising deterioration. Importance of having staff concern in a calling criteria. |
| Cioffi et al[42] | Australia | Focus groups with clinicians and nurses exploring their responses to abnormal vital signs | Constant comparison | EWS | Intuition important, particularly for more senior staff to detect AVS. Knowing your patient reported as key; knowledge and experience seen as essential. |

Continued

**Table 3** Continued

| Author | Country | Methodology | Analysis | Search area | Evidence contribution |
|---|---|---|---|---|---|
| Claussen et al[65] | USA | Retrospective review of calls to the rapid response team (RRT) and cardiac arrest calls to evaluate impact of evidence-based guidelines (preintervention). Modified Early Warning Score and huddle implemented throughout to compare pre and post. | Descriptive statistics | Electronic systems | Professional intuition used in conjunction with the Modified Early Warning Score |
| Davies et al[53] | USA | Survey looking at barriers to rapid response system (RRS) activation | Statistical analysis | EWS | Professional intuition used over RRS activation criteria—barrier to activation |
| de Groot et al[62] | Netherlands | Retrospective patient review and semistructured interviews with professionals | Descriptive statistics and grounded theory | PEWS | Importance of professional intuition is reported. |
| Donohue et al[64] | UK | Qualitative design with critical incident technique. Semistructured interviews with nurses and the outreach team. | Thematic analysis | EWS | Importance of gut feeling and intuition in detecting deterioration |
| Downey et al[74] | UK | Narrative review | 'Patterns were identified and translated to themes, which were further refined using an iterative process.' | PEWS | Drive towards automated alerts |
| Endacott et al[43] | UK | Mixed methods case study—semistructured interviews and audit of charts | Qualitative content analysis and descriptive statistics | Observations and monitoring | Gut feeling important—interestingly particularly for nurses whereas doctors sought additional empirical evidence to back up intuition. |
| Endacott and Westley[39] | Australia | Questionnaire, in-depth interviews and observations | Content analysis and constant comparison | EWS | Intuition and anticipatory skills important in detecting deterioration |
| Entwistle[73] | USA | Editorial | N/A | Family involvement | Little evidence/no evaluations of policies or practices that encourage and support family involvement in clinical monitoring. Propose the innovative practice of interdisciplinary rounds where families are invited, and communication is directed to the patient and family. |

Continued

**Table 3** Continued

| Author | Country | Methodology | Analysis | Search area | Evidence contribution |
|---|---|---|---|---|---|
| Fagan et al[82] | USA | Observational cohort comparison study | Descriptive statistics | Electronic systems | Concerns about overburdening staff with automated alerts. When considering the patient's baseline, nursing staff are expected to notify the patient's care provider immediately responsible for the patient when the vital signs meet or exceed the clinical trigger. |
| Graedon and Graedon[72] | USA focus | Opinion piece | N/A | Family involvement | Parents to explain child's baseline. Need to have detailed information about signs and symptoms that they should look for that would require alerting the medical team. Propose the need for structured inclusion of family concern during ward rounds (safety walkrounds). |
| Hands et al[44] | UK | The vital signs (VS) and early warning data for all inpatients for 1 year to investigate patterns of VS observations collected | Statistical | Observations and monitoring | Staffing levels and availability of equipment impede the collection of vital signs and early warning data. Night-time factors also play a role with a decrease in observations noted. |
| Hope et al[61] | UK | Semistructured interviews with 17 registered nurses | Constant comparative method informed by grounded theory | Snowball sample | Wider context of night-time care considered; there is some indication that staff are making a choice and prioritising sleep over monitoring. |
| James et al[37] | UK | Postal survey with healthcare assistants (HCA) using closed and open questions | Descriptive statistics and content analysis of qualitative data | Observations and monitoring | Factors other than the score used to detect deterioration—intuition/patient's own descriptions. Vital signs monitoring delegated to junior staff who may not have the skills to interpret results; many HCAs did not 'fully understand this neurological assessment and it is implications for the acutely unwell patient thus resulting in the risk of an inaccurate Early warning score' (p 552). |
| Jensen et al[46] | Denmark | Focus group exploring nurses' experiences with paediatric early warning scores | Qualitative meaning condensation analysis | PEWS | Sleeping as a part of care identified. Professional intuition. Night-time pressures. |

Continued

**Table 3** Continued

| Author | Country | Methodology | Analysis | Search area | Evidence contribution |
|---|---|---|---|---|---|
| Jones et al[66] | Australia | Questionnaire about understandings and barriers to activating a MET | Simple descriptive statistics | EWS | Intuition important when detecting deterioration |
| Jones et al[77] | UK | Electronic capture of physiological data to see if automated clinical alerts increase compliance with an early warning score and improve patient outcomes | Statistical | Electronic systems | Electronic systems can provide prompts or alerts for monitoring vital signs. This system demonstrated significant improvement in clinician attendance to patients who are acutely ill with an early warning score of 3 and above. |
| Kaul et al[24] | USA | Descriptive cross-sectional study; nurse and medical staff survey | Descriptive statistics | PEWS | Benefits of Bedside PEWS—nurses more likely to identify early signs of deterioration and a greater ability to escalate care. Delegation to junior staff justified. |
| Lobos et al[25] | Canada | Implementation report | Simple descriptive statistics | PEWS | Delegation to junior staff. Education package developed around the RRS with a variety of informal lectures and interactive sessions; 'lunch and learns', 'education coffee carts' and simulation programmes for instance. |
| Lydon et al[40] | Ireland | 30 semistructured interviews based on the theory of planned behaviour were conducted. Questionnaire with Likert scale developed from the interview data. | Deductive content analysis | PEWS | Professional intuition |
| Mackintosh et al[38] | UK | Ethnographic perspective; observations, semistructured interviews | Data were inductively and deductively coded using NVivo V.8 and organised thematically. | EWS | Delegation of routine observation and vital signs monitoring to junior staff. Professional intuition important; 'knowing the patient' important for detecting subtle changes in a patient's condition. |

Continued

**Table 3** Continued

| Author | Country | Methodology | Analysis | Search area | Evidence contribution |
|---|---|---|---|---|---|
| Mackintosh et al[27] | UK | Comparative case study—an RRS using ethnographic methods including observations, interviews and documentary review | Inductive and deductive coding facilitated by NVivo. Also used theme building and structuring methods from framework analysis while also informed by other theoretical frameworks such as 'technology-in-practice'. | EWS | Observations and monitoring delegated to junior staff/HCAs and nurses—early warning system (EWS) legitimised delegation of these tasks. Importance of pattern recognition and intuition. The TTTs used to confirm or identify deterioration and are particularly useful for junior staff. |
| Mackintosh et al[52] | UK | Ethnographic study using observations (>120 hours), semistructured interviews (n=45) and documentary review | Thematic analysis with data analysed iteratively in addition to a more strategic and policy-focused coding framework | EWS | Professional intuition important when detecting deterioration |
| Massey et al[67] | Australia | In-depth semistructured interviews | Inductive approach—thematic analysis | EWS | Importance of intuition or 'sensing clinical deterioration' |
| McDonnell et al[26] | UK | Single-centre, mixed methods, before-and-after study including a survey to measure changes in nurses' knowledge after implementation of a track and trigger system (T&Ts). Also, qualitative interviews. | Statistical analysis and thematic framework analysis | EWS | T&Ts used to confirm or identify deterioration depending on the experience of the user. Specific objective information was seen as helping the response arm prioritise work Importance of professional intuition or 'gut feeling' in detecting deterioration. Need for staff concern in T&Ts. |
| McKay et al[59] | Australia | Prospective, controlled, before-and-after intervention trial | Statistical analysis | PEWS | Specific education package developed around the paediatric early warning score which assists in the recognition of deterioration |
| Mohammed et al[75] | UK | Three phases; the first two were based in a classroom and asked nurses to calculate an early warning score from vignettes using pen and paper followed by a handheld computer. The third phase followed the previous approach but was based on the ward after nurses had been using the device for 4 weeks. | Statistical | Electronic systems | Timeliness of electronic vital signs recording when compared with paper systems |

**Table 3** Continued

| Author | Country | Methodology | Analysis | Search area | Evidence contribution |
|---|---|---|---|---|---|
| Mohammed Iddrisu et al[57] | Australia | To explore nurses' role in recognising and responding to deteriorating postoperative patients through focus groups | Thematic analysis | Snowball sample | Staff concern in tool criteria useful |
| Paciotti et al[71] | USA | Semistructured interviews with clinicians | Grounded theory and constant comparison | Family involvement | Physicians depend on families to explain a child's baseline. |
| Pattison and Eastham[29] | UK | Mixed methods study looking at the impact of a critical care outreach team (CCOT) | Statistical analysis and grounded theory | EWS | Track and trigger tool is used to either confirm or identify deterioration depending on the experience of the user. 'Gut feeling'/intuition important—patient appearance often an important sign in detecting deterioration. |
| Radeschi et al[60] | Italy | Multicentre survey to identify the attitudes and barriers to MET utilisation | Statistical analysis | EWS | Intuition (physical appearance important) in detecting deterioration |
| Schmidt et al[78] | UK | Retrospective analysis of data collected routinely. Pragmatic observational study of VitalPAC deployment in two large hospitals. | Statistical analyses | Electronic systems | Electronic systems provide prompts for alerts, facilitate better recognition of deterioration and are associated with reduced mortality. |
| Sefton et al[76] | UK | Controlled study of vital signs documentation and paediatric early warning Scorecalculation and a survey of acceptability | Descriptive statistics | PEWS | Errors in paper-based documentation were found; incorrect age-specific chart used; inaccurate documentation of values and plotting of trends; incorrect score calculation. In comparison, the error rate using the electronic score was low. |
| Shearer et al[31] | Australia | A mixed methods study | Iterative coding | EWS | Importance of professional intuition |
| Sønning et al[41] | Norway | Questionnaire of a sample of staff who use a paediatric early warning score | Descriptive statistics | PEWS | Appreciate that a paediatric early warning score encourages a systematic approach to monitoring. |
| Stevenson and Nilsson[85] | Sweden | Qualitative; focus groups with 21 registered nurses | Content analysis of interviews | Electronic systems | Verbal reports were favoured over the electronic system. |
| Stewart et al[32] | Sweden | Mixed methods. Retrospective review of records and nurse-led focus groups. | Statistical analysis and content analysis | EWS | Intuition still plays a part—the 'score' was rarely the single determining factor in activation but rather prompted nurses to gather additional information. |

Continued

**Table 3** Continued

| Author | Country | Methodology | Analysis | Search area | Evidence contribution |
|---|---|---|---|---|---|
| Subbe et al[80] | UK | A before-and-after study of an electronic automated advisory VS monitoring and notification system. Elevated scores were relayed to an RRT. | Statistical analysis | Snowball sample | Automated versus monitoring associated with a decrease in mortality |
| Wager et al[84] | USA | Observational study | Descriptive statistics | Electronic systems | Batching of patient data whereby the care provider handwrites the patient's vital signs and uploads it to the computer at a later time is common, especially as the computers are often busy. Individual tablet PCs seem to eliminate this from occurring. |
| Watson et al[45] | USA | Mixed methods, retrospective medical record observations and observations of nurse interactions in 1 min blocks | Observation analysis, although this is not described, and statistical analysis | PEWS | Information relating to transfer to electronic systems—distance the 'eyes' of the nurses from the patient; batching. Intuition. |
| Wheatley[34] | UK | Ethnographic approach; participant observation and semistructured interviews | Thematic and content analysis | Observations and monitoring | The regularity of vital signs monitoring may also lead to it becoming viewed as mundane practice that can be delegated to healthcare assistants. Workplace pressure/equipment failures affect quality of observations. |

AVS, abnormal vital signs; N/A, not applicable; TTT, track and trigger tool.

**Table 4** Summary of planning evidence

| Author | Country | Methodology | Analysis | Search area | Evidence contribution |
|---|---|---|---|---|---|
| Abraham et al[91] | USA | Pre/postprospective study | The quality and completeness of the handoff note—both tools—was assessed by a multiprofessional round. | Structured handover | Value of a checklist tool for handover |
| Brady and Goldenhar[63] | USA. | Focus groups ×7—held in groups of participants with similar roles | Constant comparison | Situational awareness | Huddle useful to proactively identify and plan for risk. Structure to support handover important. |
| Brady et al[88] | USA | | Statistical process control charts | Situational awareness | Huddles led by a watchstander charge nurse. When risk is identified the team discussed this and developed a plan to mitigate risk. |
| Claussen et al[65] | USA | Retrospective review of calls to the rapid response team (RRT) and cardiac arrest calls to evaluate impact of evidence-based guidelines | Descriptive statistics | Electronic systems | The huddle was seen as useful and was called as a patient's status was changing to 'red' so that all team members were informed. |
| Davies et al[53] | USA | Survey looking at barriers to rapid response system (RRS) activation | Statistical analysis | EWS | Activation criteria displayed around the hospital |
| Demmel et al[58] | USA | Discussion of the set-up and implementation of a paediatric early warning scoring tool and an associated algorithm | Rapid Plan-Do-Study-Act (PDSA) cycles were implemented using small tests of change. The data from the PDSA cycles were continuously collected, analysed and reviewed with the multidisciplinary staff and planning team and used to give ongoing direction to the implementation plan. | PEWS | Importance of common information spaces and display of activation criteria throughout the hospital |

**Table 4** Continued

| Author | Country | Methodology | Analysis | Search area | Evidence contribution |
|---|---|---|---|---|---|
| Donahue et al[92] | USA | Focus group evaluation of a training programme which was developed to teach paraprofessionals Situation-Background-Assessment-Recommendation (SBAR) communication tool | Not clear | Structured handover | Value of a structured approach to handover. Adaptation of the SBAR tool for handover. |
| Ennis[23] | Ireland | Description of implementation of paediatric early warning score and subsequent audit (prospective cohort observational study) | Simple descriptive statistics of numbers of children triggering the paediatric early warning score and compliance with escalation protocol | PEWS | Common information spaces important and display of activation criteria throughout the hospital. Usefulness of ISBAR as a communication tool. |
| Goldenhar et al[87] | USA | Semistructured interviews and focus groups to develop a deeper understanding of a newly implemented huddle system | Constant comparison | Situational awareness | Importance of the huddle—empowerment and sense of community; facilitated greater and better information sharing. Each huddle participant was asked to systematically report on patients in their units who they thought would deteriorate in the near future and to label them as 'watchers', asking senior nurses and physician leads to coach charge nurses on how to integrate their perceptions into an informal severity of illness assessment (comprehension) and training the clinicians on how to use the information to facilitate prediction and planning for at-risk patients (projection). |

Continued

**Table 4** Continued

| Author | Country | Methodology | Analysis | Search area | Evidence contribution |
|---|---|---|---|---|---|
| Mackintosh et al[47] | UK | Ethnographic two-stage process of lightly structured observations followed by a more focused period of data generation to test and elaborate the outcomes from stage 1 | Initial thematic analysis and search for negative cases | Situational awareness | For teams with a strong level of situation awareness (SA), key supports were all used in a balanced and supportive manner to gather and disseminate information which served to promote a culture of shared ownership and a proactive model of workload management, with temporary disruptions to the model easily accommodated. Whiteboard—when used effectively was a vital means for supporting SA. It provides teams with a snapshot of a constantly changing workload, the team's activity levels and resource availability. Workload at times compromised its effectiveness. And irrespective of workloads, location and local traditions had a bearing on its use and usefulness. Whiteboards need to be in a visible location. Importantly, the interplay between these key supports for SA will vary depending on the context; the same supports used differently will naturally produce different outcomes. Handover took many forms: structured and informal; profession specific and interprofessional; participatory or a one-way transmission of information. The effectiveness of SA was dependent on the form that handover took—who was present, contributions made, information relayed. SA was more likely to be compromised when key people were absent. Senior staff coordinator important for SA and became compromised if they were largely focused on providing patient care. |
| Mackintosh et al[27] | UK | Comparative case study—an RRS using ethnographic methods including observations, interviews and documentary review | Inductive and deductive coding facilitated by NVivo. Also used theme building and structuring methods from framework analysis while also informed by other theoretical frameworks such as 'technology-in-practice'. | EWS | Availability of equipment was an issue where the track and trigger tool (TTT) was electronic. Electronic systems helped HCAs and nursing staff to share understandings, planning and manage their workload. |
| Massey et al[67] | Australia | In-depth semistructured interviews | Inductive approach—thematic analysis | EWS | Common information spaces useful. Display of activation criteria throughout hospital. |

Continued

**Table 4** Continued

| Author | Country | Methodology | Analysis | Search area | Evidence contribution |
|---|---|---|---|---|---|
| McCrory et al[54] | USA | Prospective, preinterventional and postinterventional study to evaluate the educational intervention of teaching ABC–SBAR (airway, breathing, circulation followed by situation, background, assessment, and recommendation) | Two blinded reviewers assessed 52 video-recorded handoffs for inclusion, order and elapsed time to essential handoff information using a scoring tool. | Structured handover | Information sharing for handovers is of variable quality. A more structured approach will improve information sharing and therefore situational awareness—'without a structured hand-off tool, paediatric interns overemphasise background information and leave the reason for the call delayed'. Adaptation of the SBAR tool to include ABC—usefulness of this. |
| Mullan et al[93] | USA | Descriptive observational study | Checklists were evaluated for rates of use, completion and identification of potential safety events. | Situational awareness | The value of more structured approach to information sharing and situational awareness. Uses a checklist handover system for physicians. Checklist items focused on the status of the patient, ED providers and hospital resources. A 'Read-Do' format was designed. |
| Parker et al[89] | USA | Manual review of all eligible patient records | Descriptive statistics | PEWS | Example of a bundle around the 'watcher' category with five components that needed to be completed within 2 hours of a patient being designated as such. |
| Pearson and Duncan[30] | UK | Brief review of the evidence base surrounding paediatric early warning scores | N/A | PEWS | Value of a more structured approach to communication—advocate the use of a shared communication model, such as SBAR, to communicate findings to superiors. |
| Pezzolesi et al[90] | UK | Delphi study for tool development | Descriptive statistics. Handovers were analysed and rated according to a measure of essential skills. | Situational awareness | Information on handover—can be of variable quality. Most handover tools are adaptations of SBAR) communication tool. Important to remember that handover facilitates teamwork and this hinges on communicative forms that extend beyond the information transfer that is typically the focus of structured handover tools. |
| Randhawa et al[55] | USA | Description of the implementation process with cardiopulmonary arrest (CPA) statistics before and after implementation | Once a cycle from the implementation has been completed this is evaluated and then another cycle begins. | PEWS | Whiteboards placed in a central location displayed scores of all patients so that staff could quickly assess which patients were at high risk for deterioration. Activation criteria displayed throughout the hospital. |

Continued

**Table 4** Continued

| Author | Country | Methodology | Analysis | Search area | Evidence contribution |
|---|---|---|---|---|---|
| Stewart et al[32] | Sweden | Mixed methods. Retrospective review of records and nurse-led focus groups. | Statistical analysis and content analysis | EWS | Huddle—the score was used during the daily bed 'huddle' with the nursing administrators to evaluate current unit acuity, determine staffing need and prepare for any transfers. |
| Van Voorhis and Willis[33] | USA | Discussion paper highlighting the process of developing a paediatric RRS | N/A | PEWS | Display of activation criteria throughout the hospital on lanyards and use of whiteboards useful |
| de Vries et al[86] | Netherlands | Semistructured interview | Qualitative content analysis | PEWS | Paediatric early warning score/TTT is used to support situational awareness. Use of a score enables clinicians to have a 'bird's-eye' view over admitted patients. |
| Wager et al[84] | USA | Observational study. Explored the timeliness and quality of vital signs data entered by three different recording methods. | Descriptive statistics | Electronic systems | Batching of patient data whereby the care provider handwrites the patient's vital signs and uploads it to the computer at a later time is common, especially as the computers are often busy. |
| Watson et al[45] | USA | Mixed methods, retrospective medical record observations and observations of nurse interactions | Observation analysis, although this is not described, and statistical analysis | PEWS | Availability of equipment is a factor. |
| Weiss et al[94] | Canada | A randomised controlled trial in an academic paediatric intensive care unit (PICU) of 20 handover events | Differences between intervention and control groups were assessed using the Mann–Whitney test and multivariate linear regression. | Structured handover | Value of a more structured approach to support information sharing—a cognitive aid to facilitate handover that prompted residents to transmit this information. The handover aid was not linked to hospital information systems—so this had to be completed by hand before handover. Handover is an opportunity for learning and professional socialisation. |
| Wong et al[95] | UK | Description of user-focused design process for use of electronic monitoring and numbers of observations taken using the system. Acceptability questionnaire. | Descriptive statistics on the number of observations recorded using the SEND system and the number of active users | Electronic systems | Development of a flexible electronic system which enabled staff to have an overview of patients—reflections on disconnection |

ED, emergency department; EWS, early warning system; HCA, healthcare assistant; N/A, not applicable.

enabling individual-level clinical data to be synthesised, made sense of and shared.[19–29 33 39 41 42 46 48 50 51 56 62 66 74 86] One study, however, found that TTTs were regarded as a nursing tool and were therefore not valued by clinicians. Consequently, nurses encountered difficulties in summoning a response.[46]

Several studies also report on the use of SBAR in this context. Like TTTs, SBAR translates information into a form that provides structure, consistency and predictability when presenting patient information. SBAR has been shown to help establish common language and expectations, minimising differences in training, experience and hierarchy and facilitating nurse–clinician communication. While several papers advocate combining SBAR with TTTs,[23 25 27 30 35 45 50] none specifically evaluated SBAR use. Mackintosh *et al* highlight that audit data suggest resistance to SBAR, with others cautioning that overextending SBAR use carries the risk of SBAR fatigue and attenuation of its effects.[27]

Structured communication tools like TTTs and SBAR do not solve all the challenges of acting in response to evidence of deterioration. Barriers to action were widely reported in the literature where these tools were in place. These include: a general disinclination to seek help,[19–22 25 27 29 31 36–39 42 48 50 51 56 64 67] concerns about appearing inadequate in front of colleagues[20 22 36 38 50 67] and failure of staff to invest in the escalation or calling criteria.[21 22 49] A number of papers also reported negative attitudes to rapid response team (RRT) or medical emergency team (MET) use in the efferent component of safety systems. METs and RRTs operate outside the immediate medical team and create different issues in paediatric warning systems than when the escalation response is managed by the treating team. These include a reluctance to activate because of the perceived busyness of paediatric intensive care unit or medical staff,[20 29 39 48 50 51] because previous expectations about an appropriate response were not met, or a sense that the situation was under control (particularly when the physiological instability is in the area of expertise of the treating team).[22 29 31 38 42 50 52 64]

No literature reported on successful interventions to facilitate RRT use, but several propose strategies to support escalation where there was no designated response team in place in the efferent component. These include informal peer support, where inexperienced staff team up with more experienced staff[21 29 50 64 67]; clear structures to support action and a supportive culture that does not penalise individual decision-making, including the use of a 'no false alarms' policy so staff are not deterred from escalating care.[21 29 36 73] Senior leadership is consistently identified as important[8 20–23 25 27 30 32 33 35 47 52 58 66 67]; lack of support from superiors meant that staff are less likely to escalate and more likely to adhere to hierarchies within the current system.[25 40 66] There is some evidence to suggest that any escalation policy should be linked to an administrative arm that reinforces the system, measures outcomes and works to ensure an effective system.[27 30]

There is a small literature on family involvement in the Action subsystem. Several studies report on Condition-Help, a programme developed in the USA to support families to directly activate an RRT if they have concerns about their child's condition. Families are also becoming increasingly recognised as playing a key role in the activation of RRTs in Australia.[97] Research has evaluated the appropriateness of calls that were made by patients or relatives[33 97–101] but has not considered why calls were *not* made.[70] Involving family members in escalation demands vigilance, requiring them to take a proactive and interactive role with staff with potentially some degree of confrontation, particularly if challenging the appropriateness of decisions taken.[73 97] Families need both cognitive and emotional resources to raise concerns that involve negotiating hierarchies and boundaries.[35 70] The literature points to a degree of professional resistance to family involvement in activation, with reports of physician concern that their role would be undermined, that resources would be stretched with an increase in calls and that it might divert attention away from those in need[71 97 99 102 103] although these fears are not supported by the evidence.[71 102 104] See table 5 for a summary of the evidence relating to the action component of the model.

## Synthesis and model development

The literature in this field is heterogeneous and stronger on the sociomaterial barriers to successful afferent component paediatric early warning systems than it is on solutions. While a number of different single interventions have been proposed and some have been evaluated, there is limited evidence to recommend their use beyond the specific clinical contexts described in the papers. This reflects both the weight and quality of the evidence, the extent to which paediatric systems are conditioned by the local clinical context and also the need to attend to the relationship between system components and interventions which work in concert not in isolation. There is also a growing realisation in the quality improvement field that an intervention that has been successful in one context does not necessarily produce the same results elsewhere which cautions against a 'one size fits all' approach.[105 106]

While it is not possible to make empirical recommendations for practice, a hermeneutic review methodology enabled the generation of theoretical inferences about the core components of an optimal paediatric early warning system. These model components are logical inferences derived from an overall synthesis of the evidence, informed by our theoretical framework and clinical expertise. These are presented as a propositional model conceptualised as three subsystems: detection, planning and action (see table 1).

## DISCUSSION

This paper reports on one of three linked reviews undertaken as part of a wider UK study commissioned to develop and evaluate an evidence-based national paediatric early

**Table 5** Summary of action evidence

| Author | Country | Methodology | Analysis | Search area | Evidence contribution |
|---|---|---|---|---|---|
| Adelstein et al[49] | Australia | Prospective comparison of rapid response team (RRT) criteria breaches | Statistical | EWS | Day/night differences in activation identified. Nurses may not understand what is required for activation. Staff not investing in calling criteria. |
| Almblad et al[8] | Sweden | Retrospective review of the electronic patient record and a context assessment of the work environment using the Alberta Context Tool | Statistical | Snowball sample | Senior leadership consistently identified as important. |
| Andrews and Waterman[19] | UK | Interviews and observations | Grounded theory | EWS | Track and trigger tools (TTT) act as prompts to action. TTT used to overcome challenges in communication and particularly valuable for junior staff. Negative attitude towards calling for help. |
| Astroth et al[50] | USA | Semistructured interviews with nurses | Coding categories were generated from the data, and consensus on final themes was achieved through an iterative process. | EWS | Situation under control—no need to escalate or perceived business of medical staff discouraged staff from RRT activation. Staff encouraged to use their intuition when activating the RRT. Concern about feeling inadequate in front of colleagues a barrier to RRT activation. Inexperienced staff teaming up—led to staff trusting their own judgement. Traditional hierarchies a barrier to RRT activation—nurses more likely to call the attending physician rather than activate the RRT. |
| Azzopardi et al[20] | Australia | Survey | Statistical analysis | PEWS | Score rarely the determining factor in escalation—would not escalate for a patient who looked well but would escalate for a patient they were worried about even if not triggering. Negative attitude towards calling for help—feeling inadequate/perceived business of paediatric intensive care unit had an impact on doctors escalating but not nurses. Senior leadership is important when implementing a MET. |

Continued

**Table 5** Continued

| Author | Country | Methodology | Analysis | Search area | Evidence contribution |
|---|---|---|---|---|---|
| Bavare et al[104] | USA | Retrospective observational study of rapid response events | Descriptive statistics | PEWS | All family-activated RRT had appropriate clinical triggers with the most common being uncontrolled pain. More than half of Family-Initiated pediatric rapid response (FIRR) had a vital signs change that should have qualified clinician-RRT activation. Seventy-six per cent FIRRs needed at least one or more interventions. Twenty-seven per cent of family-initiated RRTs needed transfer to intensive care unit compared with 60% transfer rate for clinician RRTs. |
| Bogert et al[98] | USA | Implementation of Condition Help (ConditionH) | Descriptive statistics | Family involvement | Implementation of ConditionH. ConditionH being addressed during daily rounds. |
| Bonafide et al[21] | USA | Semistructured interviews | Grounded theory | PEWS | Disinclination to seek help and concerns about appearing inadequate in front of colleagues. Informal peer support. Senior leadership important. |
| Braaten[22] | USA | Document review and interviews using the principles of cognitive work analysis | Inductive and deductive forms of analysis—cognitive work analysis, framework and directed content analysis | EWS | Issues around availability of equipment and staffing. Negative attitude/delays around calling for help with staff needing to justify escalation. Other factors impact on this including the perception that the situation is under control/perceived business of physicians/not wanting to appear inadequate. |
| Brady et al[88] | USA. | Statistical process control charts | Statistical process control charts | Situational awareness | Concerns about resources reported |
| Brady et al[102] | USA | A retrospective cohort study looking at the association between family and clinician activations and transfer to the intensive care unit following a MET call | Quality improvement methods and statistical process control charts were used to assess the rate of family activation of METs. | Family involvement | Direct mechanism for families to activate a MET. Concerns from clinicians about a family-activated MET overburdening the system are unfounded. |
| Chua et al[36] | Singapore | A qualitative survey using critical incident technique | Inductive content analysis | EWS | Staff felt that they had not been educated to an adequate level—training lacking. Negative attitude towards calling for help—fears of appearing inadequate. |

Continued

**Table 5** Continued

| Author | Country | Methodology | Analysis | Search area | Evidence contribution |
|--------|---------|-------------|----------|-------------|----------------------|
| Cioffi[48] | Australia | Unstructured interviews with nurses who had activated the medical emergency team (MET) | Simple code and retrieve | EWS | Reluctance to activate—doubting ability; fears of appearing inadequate; decisions made based on the perceived availability of resources/business of medical staff/time of day all had an impact on decisions to activate the MET. Importance of having staff concern in the MET criterion. |
| Cioffi[51] | Australia | Unstructured interviews | Simple code and retrieve | EWS | Importance of having staff concern in a calling criteria. Reluctance to activate—business of ward a factor. |
| Cioffi et al[42] | Australia | Focus groups with clinicians and nurses exploring their responses to abnormal vital signs | Constant comparison | EWS | Availability of equipment an issue/staffing pressures; staff unable to carry out routine monitoring that would enable the detection of abnormal vital signs (AVS)/escalation hampered because of difficulty finding the appropriate senior person.  MET criteria used to confirm or identify deterioration depending on experience. Negative attitude towards asking for help—lack of confidence questioning peers/fear of being reprimanded/feeling the situation was under control. |
| de Groot et al[62] | Netherlands | Retrospective patient review and semistructured interviews with professionals | Descriptive statistics and grounded theory | PEWS | Easily approachable nurses and physicians, as well as good communication, were considered to be vital for timely intervention in cases of clinical deterioration in paediatric patients. Facilitators for the implementation of registration ofpaediatric early warning score included the integration of scores into the electronic patient records. |
| Dean et al[99] | USA | Two-year reflection following implementation of ConditionH | Descriptive statistics | Family involvement | ConditionH criteria for activation. Concern that family-activated RRS could divert attention away from resources. Clinician involvement important. Daily 'patient rounds' involving patients and families is useful. Patients and families have access to relevant information and understand the medical information and care plans. |

**Table 5** Continued

| Author | Country | Methodology | Analysis | Search area | Evidence contribution |
|---|---|---|---|---|---|
| Demmel et al[58] | USA | Discussion of the set-up and implementation of a paediatric early warning scoring tool and an associated algorithm | Rapid Plan-Do-Study-Act (PDSA) cycles were implemented using small tests of change. | PEWS | Education package developed around the history and development of paediatric early warning scores along with the rationale for and the goals of the initiative. The scoring process was explained and how it would be integrated into routine nursing assessments; normal vital sign parameters were reviewed. Importance of common information spaces and display of activation criteria throughout the hospital. Senior lead commitment and importance of champions integral for implementation. |
| Donohue et al[64] | UK | Qualitative design with critical incident technique. Semistructured interviews with nurses and the outreach team. | Thematic analysis | EWS | Some resistance to escalation—clinicians preferring to deal with patient problems within their own team. Inexperienced staff teaming up with more experienced staff once patient deterioration was recognised. |
| Downey et al[74] | UK | Narrative review | 'Patterns were identified and translated to themes, which were further refined using an iterative process.' | PEWS | Impact on communication—packaging information. Facilitates communication across hierarchies. |
| Endacott and Westley[39] | Australia | Questionnaire, in-depth interviews and observations | Content analysis and constant comparison | EWS | Art of referral important—using the right language and suggesting actions that would be acceptable to the doctor. Availability of equipment a factor. Negative attitude towards calling for help; escalation dependent on perceived capability of medical staff. |
| Ennis[23] | Ireland | Description of implementation of paediatric early warning score and subsequent audit (prospective cohort observational study) | Simple descriptive statistics of numbers of children triggering the paediatric early warning score and compliance with escalation protocol | PEWS | Structured education and training programme on the use of Identify-Situation-Background-Assessment-Recommendation (ISBAR) and paediatric early warning score was provided and nurse manager/staff nurse in charge should review any educational requirements in completing the paediatric early warning score particularly for relief staff. Common information spaces important and display of activation criteria throughout the hospital. Usefulness of ISBAR as a communication tool. Senior lead commitment—paediatric early warning score management policy developed/senior staff promote and reinforce use of the tool |

Continued

**Table 5** Continued

| Author | Country | Methodology | Analysis | Search area | Evidence contribution |
|---|---|---|---|---|---|
| Entwistle[73] | USA | Editorial | N/A | Family involvement | Little evidence/no evaluations of policies or practices that encourage and support family involvement in clinical monitoring. Propose the innovative practice of interdisciplinary rounds where families are invited, and communication is directed to the patient and family. |
| Gerdik et al[103] | USA | Routine data collection for number of RRT calls and the result of these activations and patient/family survey relating to RRT activation | Statistical analysis | Family involvement | Direct mechanism for families to activate the RRT. Barriers to family activation highlighted, specifically professional resistance. Physician and leadership support important to overcome barriers. |
| Gill et al[97] | Australia | Commentary drawing together family-centred care concepts, the National Safety and Quality Healthcare Service (NSQHS) Standardsand the development of family-initiated care in Australia | N/A | PEWS | Family-activated RRTs now increasingly common in Australia. In the first instance, families need to be aware of the policy. Stress the importance of understanding the number and nature of the call. Reports on health professional's resistance to it. Families need vigilance to escalate care. Need resources in order to negotiate hierarchies and boundaries. |
| Greenhouse et al[100] | USA focus | Discussion about the implementation of ConditionH | Descriptive statistics | Family involvement | Appropriateness of calls is reported rather than why they are made. Note some scepticism and wariness among nurses and physicians. |
| Hueckel et al[101] | USA | Scripted family teaching about RRT activation at the time of patient admission from ConditionH | Descriptive statistics about delivery of educational programme and RRT call-out; survey testing family understanding | Family involvement | Description of Condition Help. Appropriateness of calls is reported rather than why they are made. |
| James et al[37] | UK | Postal survey with healthcare assistants (HCA) using closed and open questions | Descriptive statistics and content analysis of qualitative data | Observations and monitoring | Workload and ward distractions a barrier to activation, such as time spent locating equipment. Disinclination to seek help from senior staff/clinicians. |

Continued

**Table 5** Continued

| Author | Country | Methodology | Analysis | Search area | Evidence contribution |
|---|---|---|---|---|---|
| Jensen et al[46] | Denmark | Focus group exploring nurses' experiences with a paediatric early warning score | Qualitative meaning condensation analysis | PEWS | Paediatric early warning score as a nursing tool and therefore not valued by medic—no universal language because of this; 'when you call and say that they have a score of 5, then they don't know what 5 means' (FG2 P1). |
| Kaul et al[24] | USA | Descriptive cross-sectional study; nurse and medical staff survey | Descriptive statistics | PEWS | Noted that the score provides a 'universal language' and interdisciplinary communication |
| Lobos et al[25] | Canada | Implementation discussion | Simple descriptive statistics | PEWS | Situation-Background-Assessment-Recommendation (SBAR) helps establish a common language and guide escalated events. Negative attitude towards calling for help—traditional hierarchies a barrier to activation/concerns about communication between primary and responding team. No false alarms and debriefing useful. Importance of champions (using a social marketing approach) to encourage 'inter-professional collaboration & advisory group to help establish a sense of ownership'. Lack of support from superiors means less likely to escalate. |

Continued

**Table 5** Continued

| Author | Country | Methodology | Analysis | Search area | Evidence contribution |
|---|---|---|---|---|---|
| Mackintosh et al[27] | UK | Comparative case study—a rapid response system (RRS) using ethnographic methods including observations, interviews and documentary review | Inductive and deductive coding facilitated by NVivo. Also used theme building and structuring methods from framework analysis while also informed by other theoretical frameworks such as 'technology-in-practice'. | EWS | Availability of equipment an issue where the TTT was electronic. Gave junior staff licence to escalate care. Additionally, 'while standardisation of practice clearly has its benefits, it also comes at a cost that these tools attenuate lower level staff's authority and ability to persuade staff higher up in the organisation of the credibility of their knowledge' (p 143). Efforts to develop junior staff's communication and clinical understanding need to acknowledge power dynamics at play. Usefulness of SBAR communication tool as part of the escalation policy as reported by staff (not seen in action). Negative attitude towards escalation—difficulty in summoning a response. Senior lead commitment to patient safety was important. Zero tolerance for cardiac arrest was championed by senior staff. Night-time/out-of-hours pressures identified. |
| Mackintosh et al[38] | UK | Ethnographic perspective; observations, semistructured interviews | Data were inductively and deductively coded and organised thematically. | EWS | Negative attitude towards seeking help. Escalating care outside the parameters marked by a track and trigger tool proved difficult; power struggles identified—junior staff have difficulty persuading more senior staff of the credibility of their knowledge. Difficulties in activation across professional boundaries. |
| Massey et al[67] | Australia | In-depth semistructured interviews | Inductive approach— thematic analysis | EWS | Common information spaces useful. Display of activation criteria throughout hospital. General negative attitude towards calling for help— appearing inadequate in front of others. Importance of leadership support. Peer support—would often consult their colleagues. |

Continued

**Table 5** Continued

| Author | Country | Methodology | Analysis | Search area | Evidence contribution |
|---|---|---|---|---|---|
| McCabe et al[35] | UK | Opinion piece about lessons to be learnt from the adult experience of implementing early warning systems | N/A | PEWS | Specific education package needed on how to use an early warning system (EWS) and on basic clinical assessment, guidance and standardisation of observation and monitoring. Advocate situational simulated scenario education and e-learning. Highlight the usefulness of communication tools such as SBAR for establishing roles and responsibilities, engaging them in making an appropriate management plan that can, if necessary, be escalated. Senior lead commitment key—reflected in resources and education—to improve the safety and quality of care of hospitalised patients. Families need to be empowered to request a patient review. |
| McDonnell et al[26] | UK | Single-centre, mixed methods before-and-after study including a survey to measure changes in nurses' knowledge after implementation of a track and trigger system (T&Ts). Also, qualitative interviews. | Statistical analysis and thematic framework analysis | EWS | Rolling education programme for all nurses on the recognition and response to deteriorating patients and an overview of the T&Ts. Workplace pressures; nurses concerned that they could not always summon a timely response from doctors/night-time pressures also identified. Need for staff concern in T&Ts. |
| Monaghan[28] | UK focus | Commentary on the development of the Brighton paediatric early warning score and setting up a paediatric critical care outreach team | Simple descriptive statistics of all activations, actions and outcomes during the first 3 months of implementation | PEWS | Education-based model was developed to assist in recognising deterioration. Temporary staff/workplace pressures impact on staff's ability to detect deterioration. |

Continued

**Table 5** Continued

| Author | Country | Methodology | Analysis | Search area | Evidence contribution |
|---|---|---|---|---|---|
| Paciotti et al[71] | USA | Semistructured interviews with clinicians to explore physicians' viewpoints on families facilitating the identification of children with a deteriorating condition | Grounded theory and constant comparison | Family involvement | Concerns that resources would be diverted away with an increase in calls—not supported |
| Pattison and Eastham[29] | UK | Mixed methods study looking at the impact of a critical care outreach team (CCOT) | Statistical analysis and grounded theory | EWS | Availability of equipment an issue/workload. Negative attitude towards calling for help—situation under control/ward business. Inexperienced staff teaming up/checking with peers before calling the CCOT. |
| Pearson and Duncan[30] | UK | Brief review of the evidence base surrounding the paediatric early warning score together with reflections from their own experiences from the Birmingham Children's Hospital | N/A | PEWS | Team training and education is important increasing confidence in the use of medical language and empowering bedside carers. 'Although doing observations is fundamental to nursing practice many … have not been taught a structured approach to assessment.' Advocate a simulated environment. Value of a more structured approach to communication—advocate the use of a shared communication model such as SBAR to communicate findings to superiors. Need for senior commitment—cultural change may be required to ensure management support (reflected in resources and education)/importance of champions. |
| Salamonson et al[56] | Australia | Survey with closed and open questions to examine perceptions of and satisfaction with the MET | Descriptive statistics and content analysis | EWS | Need for more education on deterioration identified. Negative attitude towards asking for help; attitude of MET team a barrier to activation. |

Continued

**Table 5** Continued

| Author | Country | Methodology | Analysis | Search area | Evidence contribution |
|---|---|---|---|---|---|
| Shearer et al[31] | Australia | A multimethod study; a point prevalence survey; a prospective audit of all patients experiencing a cardiac arrest, unplanned intensive care unit (ICU) admission or death over an 8-week period. Structured interviews with staff to explore cognitive and sociocultural barriers to activation. | Iterative coding | EWS | Adequate staffing and a lack of beds on critical care leads to a failure to activate the RRS. Score rarely the single determining factor in activation despite the fact that staff recognised patients met activation criteria. Data from the point prevalence study confirm this as only one patient had a serious adverse event. Negative attitude towards calling for help—situation under control; treating team had expertise to treat (particularly when the physiological instability was in the area of expertise of the treating team). Traditional (intraprofessional clinical) hierarchies a barrier to activation. |
| Sønning et al[41] | Norway | Questionnaire of a sample of staff who use a paediatric early warning score | Descriptive statistics | PEWS | Nurses gain self-confidence. More effective communication. |
| Stewart et al[32] | Sweden | Mixed methods. Retrospective review of records and nurse-led focus groups. | Statistical analysis and content analysis | EWS | The RRS was valuable for junior staff escalating care across hierarchical and professional boundaries. Senior lead commitment—culture of support promoted by nursing administrators. |

Continued

**Table 5** Continued

| Author | Country | Methodology | Analysis | Search area | Evidence contribution |
|---|---|---|---|---|---|
| Van Voorhis and Willis [33] | USA | Discussion paper highlighting the process of developing a paediatric RRS. The system was evaluated by prospectively collected data recorded on RRS activation forms and existing performance improvement database information. | N/A | PEWS | Display of activation criteria throughout the hospital on lanyards and use of whiteboards useful. Debriefing following activation and a commitment to no false alarms is encouraged. Senior lead commitment—administrative arm of the RRS vital. Uses Condition Help. The appropriateness of calls was facilitated by the 'no false alarms' culture. |
| de Vries et al [86] | Netherlands | Semistructured interview | Qualitative content analysis | PEWS | Paediatric early warning score facilitated communication across hierarchies. |
| Watson et al [45] | USA | Mixed methods, retrospective medical record observations and observations of nurse interactions in 1 min blocks | Observation analysis, although this is not described, and statistical analysis | PEWS | Availability of equipment a factor. Score rarely the determining factor in escalation. SBAR. |

N/A, not applicable.

warning system.[3] Drawing on TMT and NPT, we have synthesised and analysed the findings from the review to develop a propositional model to specify the core components of optimal afferent component paediatric early warning systems. While there is a growing consensus of the need to think beyond TTTs to consider the whole system, no frameworks exist to support such an approach. Clinical teams wishing to improve rescue trajectories should take a whole systems perspective focused on the constellation of factors necessary to support detection, planning and action and consider how these relationships can be managed in their local setting. TTTs have value in paediatric early warning systems but they are not the sole solution and depend on certain preconditions for their use. An emerging literature highlights the importance of planning and indicates that combinations of interventions may facilitate situation awareness. Professional judgement is also important in detecting and acting on deterioration and the evidence points to the importance of a wider organisational culture that is supportive of this. Innovative approaches are needed to support family involvement in all aspects of paediatric early warning systems, which are sensitive to the cognitive and emotional resources this requires. System effectiveness requires attention to the sociomaterial relationships in the local context, senior support and leadership and continuous monitoring and evaluation. New technologies, such as moving from paper-based to electronic TTTs, have important implications for all three subsystems and critical consideration should be given to their wider impacts and the preconditions for their integration into practice.

## Limitations of the review

The literature in this field is heterogeneous and better at identifying system weakness than it is effective improvement interventions. It was only by deploying social theories and a hermeneutic review methodology did it prove possible to develop a propositional model of the core components of an afferent component paediatric early warning system. This model is derived from logical inferences drawing on the overall evidence synthesis, social theories and clinical expertise, rather than strong empirical evidence of single intervention effectiveness. Consequently, there is a growing consensus of the need to take a whole systems approach to improve the detection and response to deterioration in the inpatient paediatric population.

## CONCLUSION

Failure to recognise and act on signs of deterioration is an acknowledged safety concern[1] and TTTs are a common response to this problem. There is, however, a growing recognition of the importance of wider system factors on the effectiveness of responses to deterioration.[5 7] We have reviewed a wide literature and analysed this using social theories to develop a propositional model of an optimal afferent component paediatric early warning system that can be used as a framework for paediatric units to evaluate their current practices and identify areas for improvement. TTT use should be driven by the extent to which teams think that they will help improve the effectiveness of their system as a whole.

**Author affiliations**

¹Centre for Trials Research, Cardiff University, Cardiff, UK
²University Library Services, Cardiff University, Cardiff, UK
³Faculty of Health and Applied Sciences (HAS), University of the West of England Bristol, Bristol, UK
⁴Alder Hey Children's NHS Foundation Trust, Liverpool, UK
⁵Department of Pediatric Emergency Medicine, Sidra Medical and Research Center, Doha, Qatar
⁶Division of Population Medicine, School of Medicine, Cardiff University, Cardiff, UK
⁷Emergency Department, Paediatric Emergency Medicine Leicester Academic (PEMLA) Group, Leicester, UK
⁸SAPPHIRE Group, University of Leicester Department of Health Sciences, Leicester, UK
⁹School of Healthcare Sciences, Cardiff University, Cardiff, UK

**Acknowledgements** The authors acknowledge the contribution of Dr Heather Strange to the review. The authors also extend their thanks to the parental advisory group who have helped shape the broader questions of the research study as well as offer guidance on wider contextual factors to consider within the overall paediatric early warning system.

**Contributors** NJ: screening and review of papers; led the theoretical synthesis of the literature; contributed to model development; preparation and writing of the manuscript (with DA). YM: screening and review of papers; contributed to model development; contributed to the drafting of the manuscript. AL: led the model development (with DA); contributed to the drafting of the manuscript. MM: conceived and led the systematic search strategies; review of manuscript. LNT, GS, CP, DR: screening and review of papers; contributed clinical expertise; contributed to model development; contributed to the drafting of the manuscript. RT: screening and review of manuscript. KH: contributed to model development; contributed to the drafting of the manuscript. DA: conceived and designed the review; led the theoretical framing and analysis; screening and review of papers; led the model development (with AL); and led the writing of the manuscript (with NJ).

**Funding** This study is funded by the National Institute for Health Research (NIHR) Health Services and Delivery Research (HS&DR) programme (12/178/17).

**Disclaimer** The views and opinions expressed in this paper are those of the authors and not necessarily those of the NHS, the NIHR or the Department of Health.

**Competing interests** None declared.

**Patient consent for publication** Not required.

**Provenance and peer review** Not commissioned; externally peer reviewed.

**Data availability statement** All data relevant to the study are included in the article or uploaded as supplementary information.

**ORCID iDs**
Nina Jacob http://orcid.org/0000-0002-3240-4179
Yvonne Moriarty http://orcid.org/0000-0002-7608-4699

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
