## [Reviewer comments · BMJ Open]

ARTICLE DETAILS

TITLE (PROVISIONAL)	Optimising Paediatric Afferent Component Early Warning Systems: A hermeneutic systematic literature review and model development
AUTHORS	Jacob, Nina; Moriarty, Yvonne; Lloyd, Amy; Mann, Mala; Tume, Lyvonne N.; Sefton, G.; Powell, Colin; Roland, Damian; Trubey, Robert; Hood, KerENZA; Allen, Davina

VERSION 1 – REVIEW

REVIEWER	Orsola Gawronski Bambino Gesù Children's Hospital, Italy
REVIEW RETURNED	26-Mar-2019

GENERAL COMMENTS	This article addresses a very relevant topic for pediatric care, the hermeneutic research and synthesis strategy is very appropriate, clearly defined and well written. This review is an important contribution to pediatric care and the improved implementation of pediatric early warning systems.
--

REVIEWER	Stephanie R Brown, MBBS Pediatric Critical Care Fellow Seattle Children's Hospital University of Washington Seattle, WA, USA
REVIEW RETURNED	08-May-2019

GENERAL COMMENTS	The authors have chosen to address important, and often inadequately discussed, issues surrounding the implementation of PEWS in this paper. It is well written, but there were a few areas that could be improved for clarity: - Definition of the "afferent" limb of PEWS - in particular it is unclear why "action" was included as one of the three components. Some important barriers to action were stated, but explaining why action is considered part of the afferent limb would be helpful for ease of understanding.- Careful language is required in describing the afferent limb of PEWS, because there is already widespread misunderstanding of what PEWS is supposed to be and resulting misuse of PEWS (ie. as a severity of illness score, without a resulting response algorithm)- Literature was included only from high resource settings, was this intentional? Studies from resource limited settings may provide valuable insight into critical barriers to implementation. Additionally, open access journals have a broad readership and
---

	many people working in Resource Limited Settings rely on open access to keep up to date. It is acceptable to omit this area of the literature, if it does not meet the requirements defined by the study team, but should be addressed in the review. - On page 11 there is reference to evidence that these tools are less effective in some populations. However, there is also evidence that PEWS is effective across a broad range of underlying conditions, this could be presented in a more balanced manner. - Table 2: The two right hand columns don't consistently line up, making the table a little hard to follow. The green section needs a label (I assume "action"). The propositional model doesn't include any mention of calculation of a score, meeting a threshold score or a response algorithm, is this intentional? Overall, this is an interesting and well written paper. In particular, the involvement of key stakeholders in the process (ie. public and parents), the extensive review of the literature and systems based approach are important strengths.
--	--

VERSION 1 – AUTHOR RESPONSE

Definition of the “afferent” limb of PEWS – in particular it is unclear why “action” was included as one of the three components. Some important barriers to action were stated, but explaining why action is considered part of the afferent limb would be helpful for ease of understanding.	We have clarified our definition of the afferent component on page 5; that the afferent component is about both the detection of deterioration and the initiation of timely and appropriate action. The literature on the effectiveness of the efferent limb responses RRTs was deliberately excluded from the review.
Careful language is required in describing the afferent limb of PEWS, because there is already widespread misunderstanding of what PEWS is supposed to be and resulting misuse of PEWS (i.e. as severity of illness score, without a resulting response algorithm).	Thank you for highlighting this important point. We have clarified this point on page 9, where we maintain the importance of having all four components of the overall safety system (afferent, efferent, process improvement & an administrative arm) working in concert in order to uphold an optimal paediatric early warning system.
Literature was included from high resource settings, was this intentional? Studies from resource limited settings may provide valuable insight into critical barriers to implementations. Additionally, open access journal have a broad readership and many people working in Resource Limited Settings rely on open access to keep up to date. It is acceptable to omit this area of the literature, if it does not meet the requirements defined by the study team, but should be addressed in the review	We included literature from all resource settings, although excluded grey literature in order to keep the review manageable. This has been outlined on page 6.
On page 11 there is reference to evidence that these tools are less effective in some	Despite widespread use and apparent face validity of PEWS, the evidence for their

populations. However there is also evidence that PEWS is effective across a broad range of underlying conditions, this could be presented in a more balanced manner.	effectiveness is weak. Our linked systematic review into the validity of TTT found that many have been have only been validated retrospectively and post predictive value were generally low. Studies reporting significant decreases in cardiac arrest calls or mortality had methodological concerns (Trubey et al. 2019). This has been outlined on page 9.
Table 2: the two right hand columns don't consistently line up, making the table a little hard to follow. The green section needs a label (I assume "action"). The propositional model doesn't include any mention of calculation of a score, meeting a threshold or a response algorithm, is this intentional?	The propositional model includes examples of interventions that are reported in the literature that might help to operationalise the conceptual requirements that are for needed for an optimal afferent paediatric early warning system. Track and trigger tools are included as an example of achieving requirements needed in the detect and act subsystem, but it is important to point out that there may be other means that can be used to achieve the same outcome.

VERSION 2 – REVIEW

REVIEWER	Stephanie R Brown, MBBS Fellow, Pediatric Critical Care Medicine Seattle Children's Hospital University of Washington Seattle, WA, USA
REVIEW RETURNED	30-Aug-2019
GENERAL COMMENTS	Thank you for your response to the reviewer comments and for addressing the suggestions raised by the review.